# Microencapsulation of Chilean Papaya Waste Extract and Its Impact on Physicochemical and Bioactive Properties

**DOI:** 10.3390/antiox12101900

**Published:** 2023-10-23

**Authors:** Yihajara Fuentes, Claudia Giovagnoli-Vicuña, Mario Faúndez, Ady Giordano

**Affiliations:** 1Departamento de Química Inorgánica, Escuela de Química, Facultad de Química y de Farmacia, Pontificia Universidad Católica de Chile, Macul 7820436, Chile; yyfuentes@uc.cl; 2Departamento de Farmacia, Escuela de Química y Farmacia, Facultad de Química y de Farmacia, Pontificia Universidad Católica de Chile, Macul 7820436, Chile; mfaundeza@uc.cl

**Keywords:** Chilean fruit, agro byproducts, maltodextrin, freeze-drying

## Abstract

The microencapsulation of bioactive extracts of Chilean papaya waste, including both seeds and skin, was investigated. Papaya waste extract microcapsules utilizing maltodextrin at 10% (MD10), 20% (MD20), and 30% (MD30) (*w*/*v*) as the wall material through the freeze-drying process were obtained, and subsequently their physicochemical, antioxidant, and antimicrobial properties were evaluated. The TPC efficiency and yield values achieved were more than 60% for the microencapsulated seed and skin extracts, respectively. The best results for phenolic and antioxidant compounds were found in the microencapsulated seed extract with MD20, with a value of 44.20 ± 3.32 EAG/g DW for total phenols and an antioxidant capacity of 12.0 ± 0.32 mol ET/g DW for the DPPH and 236.3 ± 4.1 mol ET/g DW for the FRAP assay. In addition, the seed and skin samples reduced ROS generation in H_2_O_2_-treated Hek293 cells. In terms of antimicrobial activity, values ranging from 7 to 15 mm of inhibitory halos were found, with the maximum value corresponding to the inhibition of *S. aureus*, for both microencapsulated extracts. Therefore, the successful microencapsulation of the waste bioactive extracts (seed and skin) with the demonstrated antimicrobial and antioxidant properties highlight the bioactivity from Chilean papaya waste resources.

## 1. Introduction

Currently, the study of native South American fruits is internationally popular, especially driven by their high content of bioactive compounds beneficial to health. The Chilean papaya (*Vasconcellea pubescens*) can be recognized by its small size, yellow flesh and skin, characteristic aroma, and the variety of biologically active compounds it contains, including carotenoids, vitamins, polyphenols, and polysaccharides [1]. In contrast to tropical papaya (*Carica papaya* L.), this fruit grows in temperate areas of Chile, and it is commonly prepared and marketed as candied, canned, and in juice, syrup, and jam forms [1,2]. However, the processing of papaya fruit in the industrial sector generates a significant quantity of waste (peel and seeds), equivalent to 20–25% of the fruit’s overall weight [3].

It has been demonstrated that some industrial food waste contains the same or even more bioactive compounds as the edible parts, resulting in biological activities that positively impact health [4,5,6,7]. However, these compounds are susceptible to environmental stressors. Exposure to high temperatures, oxygen, water, and light during production, storage, and transportation can lead to a loss of biological value, bioavailability, solubility, and functionality [8]. To address this issue, microencapsulation emerges as an alternative to protect the bioactive compounds from plant matrices, reducing susceptibility and enhancing stability. Several encapsulation methods have been employed to stabilize and protect bioactive compounds, with freeze-drying being one of the most common techniques. This method is ideal for microencapsulating heat-sensitive materials due to its low temperature. As a result, freeze-drying microencapsulation preserves the biological activity of bioactive compounds and delays their degradation [9].

Maltodextrin has gained significant attention as a versatile material for microencapsulation due to its unique properties [10]. Serving as a wall material, maltodextrin, a complex carbohydrate derived from starch, offers several advantages in the microencapsulation process. For example, this wall material has excellent solubility, allowing for easy encapsulation of a wide range of hydrophobic and hydrophilic bioactive compounds [11]. The biocompatibility, polyfunctionality, and low cost of maltodextrin further contribute to its appeal as a material in the microencapsulation process [12].

In this research, we investigated the microencapsulation of bioactive extracts derived from Chilean papaya waste, which includes both the seed and skin components. Our primary objectives encompassed not only the production of bioactive microcapsules from papaya waste extracts but also an extensive analysis of their physicochemical properties. We assessed their total antioxidant capacity (TAC) using various methods, including DPPH, FRAP, and ABTS, and conducted kinetic analysis of intracellular ROS generation. Additionally, we determined the total polyphenol content (TPC) and total flavonoid content (TFC). To further enhance our understanding, we identified and quantified the phenolic compounds using LC–MS/MS analysis. The findings from this research open promising possibilities for the utilization of fruit waste in various industries, including but not limited to food, cosmetics, and pharmaceuticals, thereby highlighting its potential to transform discarded resources into valuable assets.

## 2. Materials and Methods

### 2.1. Chemicals and Reagents

The following chemicals were acquired from Sigma-Aldrich (Sigma-Aldrich, St. Louis, MO, USA): aluminum chloride, sodium acetate, sodium carbonate, Folin–Ciocâlteu reagent (FC), gallic acid, ferulic acid, chlorogenic acid, caffeic acid, coumaric acid, rutin, quercetin, DPPH (2,2-diphenyl-1-picrylhydrazyl), ABTS (2,2-Azino-bis (3-ethylbenzothiazo-line-6-sulphonic acid)), TPTZ (2,4,6-tripyridyl-s-triazine), FeCl_3_ (ferric chloride hexahydrate), ethanol, standard of quercetin, gallic acid, and Trolox (6-hydroxy-2,5,7,8-tetramethylchromane-2-carboxylic acid). The DIMEM F12, non-essential amino acids, penicillin-streptomycin, trypsin, and fetal bovine serum were obtained from Biological Industries Co. (Beit HaEmek, Israel). The HBSS, PBS, and Dichlorofluorescein diacetate were purchased from Merck (Merck, Darmstadt, Germany).

### 2.2. Collection and Preparation of Raw Material

Chilean papaya waste (skin and seeds) of the *Vasconcellea pubescens* variety was collected from a local market in La Serena city, the Coquimbo region, Chile. The skin and seeds were frozen at −80 °C for 12 h and then freeze-dried (Labconco Freezone 1L, Kansas, MO, USA) at a −40 °C condenser temperature and 0.06 mbar chamber pressure for 48 h. Then, the freeze-dried skin and seeds were grounded into a powder and stored at 25 °C in desiccators. Table 1 presents the proximate analysis of the raw materials (seed and skin) obtained from Chilean papaya waste.

### 2.3. Preparation of Chilean Papaya Waste Extracts

The extraction protocol was undertaken according to a previous body of work with modifications [13]. To each of the papaya waste powders (12 g), 70% ethanol (24 mL) was added and sonicated for 60 min using an ultrasonic bath (VWR^®^ Symphony™ Ultrasonic Cleaners, Radnor, PA, USA; 35 kHz operation frequency). The papaya waste with the extraction solvent was covered with aluminum foil and stored (4 °C) for 24 h in the dark. After centrifugation at 9520× *g* for 30 min (MSE Super minor, Fisons, Ipswich, Suffolk, UK), the supernatants were collected, filtered (0.22 μm syringe filters), and evaporated to obtain the papaya waste extracts.

### 2.4. Preparation of Microcapsules

The microencapsulation procedure by Giovagnoli-Vicuña et al. [14] was used. To prepare the microcapsules, maltodextrin was dissolved in water at different concentrations (10%, 20%, and 30% *w*/*v*). The papaya waste extracts of skin and seeds were then added so that the ratio of waste extracts to wall material was 1:2. The mixtures were homogenized in a magnetic stirrer (Scilogex MS-H280-Pro, Scilogex LLC, Rocky Hill, CT, USA) at 100 rpm for 30 min and freeze-dried (at −40 °C for 24 h).

### 2.5. Microencapsulation Yield

Microcapsule recovery (%) was defined as the mass of the obtained freeze-dried product over the percentage quantity of the initial dried extract.

### 2.6. Physicochemical Characterization

The following analyses were performed in the physicochemical characterization of the microcapsules: Moisture content, determined according to AOAC method no. 934.06 [15], and the water activity (Aw) was measured using a water activity meter (4TE, AquaLab, Pullman, WA, USA). For hygroscopicity, the samples were subjected to 75% relative humidity (NaCl saturated solution) at 25 °C for 7 days, according to Nunes et al. [16]. Solubility in water was determined using the method reported by Flores-Mancha et al. [17]. For observation of the morphology and size of the microcapsules with a scanning electronic microscopic (Hitachi SU3500, Tokyo, Japan) working with a voltage of 25 kV, the sample preparation and the observation conditions were described by Giovagnoli-Vicuña et al. [14]. The color was analyzed using a FRU WR-10 colorimeter (Weifu Photoelectric Technology Co., Ltd., Shenzhen, China), which provided CIE L*, a*, and b* values.

### 2.7. Thermal Analysis

The experimental runs were carried out using a thermo-gravimetric analyzer (TGA 4000, Perkin-Elmer Inc., Wellesley, MA, USA) [18,19]. The samples were heated from 20 °C to 500 °C at a heating rate of 10 °C min^−^^1^ under a nitrogen flow rate of 50 mL/min^−^^1^.

### 2.8. Total Phenolic (TPC) and Total Flavonoid Contents (TFC)

To determine the TPC and TFC in the microcapsules, we followed the methodology proposed by Saikia et al. [19] with some modifications. We took 100 mg of the microencapsulated sample and added 1 mL of a mixture containing ethanol, acetic acid, and water in a ratio of 50:8:42 (*v*/*v*/*v*). This mixture was vortexed for 1 min at room temperature, followed by 20 min using an ultrasonic bath (VWR^®^ Symphony™ Ultrasonic Cleaners, Radnor, PA, USA; 35 kHz operation frequency). The resulting supernatant was then centrifuged (ORTO ALRESA, Bioprocen 22 R model) for 5 min at 112× *g* and filtered. Finally, the supernatant obtained after filtration was subjected to sample concentration (model MD200-1A, ALLSHENG Instrument Co., Ltd., Hangzhou, China) until complete dry, and the volume was reconstituted with 70% ethanol.

The TPC was determined using the Folin–Ciocâlteu (FC) methodology [20]. Specifically, 20 µL of the extract was mixed with 100 µL of the FC reagent at a 1:10 (*v*/*v*) ratio and combined with 80 µL of a 7.5% sodium carbonate solution. The resulting mixture was then allowed to react for 40 min in darkness before measuring the absorbance at 765 nm (FlexA-200 microplate reader, ALLSHENG Instrument Co., Ltd., Hangzhou, China). The TPC was calculated using a calibration curve (10–100 µg/mL; y = −0.0048x + 0.0026; R^2^ = 0.9946) with gallic acid (GA) as the standard and expressed in milligrams of gallic acid equivalent (GAE) per gram of dry weight of the sample (DW).

The total flavonoid content (TFC) was determined using the aluminum chloride procedure [21]. To accomplish this, 100 microliters of the samples were mixed with 100 µL of a 2% AlCl_3_ ethanol solution and incubated for 30 min at room temperature. Subsequently, the absorbance was measured at 420 nm using a microplate reader (FlexA-200, ALLSHENG Instrument Co., Ltd., Hangzhou, China). The TFC was calculated using a calibration curve (10–90 µg/mL; y = −0.0095x + 0.0294; R^2^ = 0.9969) with quercetin as the standard and expressed in milligrams of quercetin equivalent (QE) per gram of dry weight of the sample (DW).

### 2.9. Microencapsulation Efficiency (MEE)

Microencapsulation efficiency (%) was determined according to Saikia et al. [19], based on the total content of bioactive compounds and the surface bioactive content. The microencapsulation efficiency (ME) was determined according to the following equation:$$ME\left(\%\right)=\left(\frac{TPC-{TPC}_{S}}{TPC}\right)\times 100$$
where the TPC is the phenolic content inside the core of the microencapsulate; TPC_S_ is the surface phenolic content.

### 2.10. Identification and Quantification of Phenolic Compounds

The identification and quantification of polyphenolics were conducted according to Velásquez et al.’s [22] methodology. All the samples were analyzed on an ultra-high pressure liquid chromatograph (Ekspert Ultra LC 100-XL system, Eksigent Technologies, Dublin, CA, USA) coupled to a triple quadrupole mass spectrometer in positive electrospray mode (ESI) (AB Sciex Triple Quad 4500, Framingham, MA, USA). A LiChrospher 100 RP-18 endcapped column (125 mm × 4 mm id, 5 µm) (Merck, Darmstadt, Germany) at 30 °C was employed for chromatographic separation with a mobile phase of 0.1% formic acid and methanol in gradient mode at a 0.5 mL min^−1^ flow rate. The gradient was programmed as follows: 0–1 min, 15% B; 1–17 min, 15–100% B; 17–21 min 100–100% B; 21–22 min, 100–15% B; 22–25 min, 15–15% B. The LC–MS/MS system was controlled using Analyst 1.6.2 and the data were processed using Multiquant 3.0. For the quantitative analysis of phenolic compounds, commercial standards (see Section 2.1) were employed. The analysis involved the use of a primary transition for quantification and a secondary transition for identification purposes (For more details, please refer to Appendix A in Appendix A.).

### 2.11. Measurement of Antioxidant Capacity

The antioxidant capacity was measured using the 2,2′-diphenyl-1-picryhydrazyl free radical scavenging (DPPH) [23], 2,2′-azino-bis (3-ethylbenzothiazoline-6-sulphonic acid) radical scavenging (ABTS^•+^) [24], and ferric reduction antioxidant power (FRAP) [25] assays. For each measurement, we combined 50 µL of the extract with 150 µL of the respective solution (DPPH, ABTS, or FRAP). After a 30 min incubation period, we recorded the absorbance at 517 nm for DPPH, at 732 nm for ABTS, and at 593 nm for FRAP, using a microplate reader (FlexA-200, ALLSHENG Instrument Co., Ltd., Hangzhou, China). Trolox served as the calibration standard. The DPPH assay produced a linear calibration curve in the range of 10–80 µg/mL (y = −0.0101x + 0.7881; R^2^ = 0.9929), while the ABTS assay exhibited a linear relationship in the range of 2–80 µg/mL (y = −0.0103x + 0.8170; R^2^ = 0.9966). The FRAP assay, on the other hand, demonstrated linearity within the range of 2–35 µg/mL (y = −0.0259x + 0.0481; R^2^ = 0.9985). The results are reported as the µmol of Trolox equivalents (ET) per gram of dry weight of the sample (DW).

### 2.12. Cell Culture and Analysis of Intracellular ROS Generation

Hek293 cells were maintained in DMEM F12 medium. Media was supplemented with Fetal Bovine Serum (5% for assays and 10% for growth), 1× non-essential amino acids and 100 U/mL penicillin and 100 μg/mL streptomycin. All cells were cultured in a humidified incubator at 5% CO_2_ atmosphere and 37 °C (NuAire, NU-5700, Plymouth, Devon, UK).

Determination of intracellular ROS generation was performed according to Gallardo-Garrido et al. [26]. The Hek293 cells were incubated with DCFDA 200 μM in HBSS. After incubation (30 min at 37 °C), the cells were washed twice with sterile HBSS and then seeded at 5 × 10^4^ cells per well, and extracts were added (1 μL). Hydrogen peroxide (H_2_O_2_) 200 μM was used for ROS generation control. Fluorescence generation was measured every 2 min for 1 h with excitation/emission wavelengths of 480/520, respectively, in a Cytation5 plate reader (Biotek^®^, Winooski, VT, USA). The results are expressed as fold changes in fluorescence generation under each condition over time.

### 2.13. Measurement of Antimicrobial Activity

The antimicrobial activity was determined using the agar diffusion method and minimum inhibitory concentration (MIC) [22]. The bacteria strains evaluated were *Escherichia coli* (ATCC-25922), *Salmonella thyphi* (ATCC-700623), and *Staphylococcus aureus* (ATCC-25923).

### 2.14. Statistical Analysis

All the data were analyzed using Statgraphics Plus^®^ 5.1 software (Statgraphics Technologies Inc., The Plains, VA, USA) and expressed as the mean ± standard deviation. Statistical comparisons were made using analysis of variance (*p* < 0.05), followed by Fisher’s test and the multiple range test.

## 3. Results and Discussions

### 3.1. Microencapsulation Yield

Three maltodextrin concentrations (10%, 20%, and 30% *w*/*w*), respectively, were studied on the microencapsulation yield from the papaya waste extract (seeds and skin; Figure 1). The results showed that maltodextrin at 20% exhibited the highest microencapsulation yield (MD20; 91.7 ± 2.2%), followed by maltodextrin at 10% (MD10; 89.9 ± 1.4%) and maltodextrin at 30% (MD30; 78.5 ± 2.6%) for the seed extract. However, there was no significant difference (*p* > 0.05) in the microencapsulation yield between MD10 and MD20. While the microcapsules of the skin extract produced with MD10 presented the highest microencapsulation yield (90.7 ± 1.0%), followed by MD30 (84.7 ± 3.4%) and MD20 (76.8 ± 4.2%), there was no significant difference (*p* > 0.05) between MD20 and MD30. Therefore, the microencapsulation yield values of the seed and skin extracts were >50%, resulting in a successful process according to what was reported by Balci-Torun and Ozdemir [27].

### 3.2. Physicochemical Characterization and Thermal Analysis

Moisture content and water activity are two variables related to microbial growth in dry foods. These parameters must be kept below 10% for moisture content and 0.60 for water activity to avoid microbial growth [28]. The extract showed the highest moisture and water activity values, indicating that its stability could be affected (Table 2). The moisture content of the microcapsules produced from seed and skin extracts was within the acceptable range of <10% [29], in line with results from other research [30,31,32,33]. Meanwhile, aw values in the microencapsulated extracts of the seed and skin of papaya waste ranged between 0.282 ± 0.003 and 0.368 ± 0.012. Likewise, some studies have reported aw values of microcapsules < 0.5, which indicate the high stability of microcapsules against microbial growth [29,34,35].

The capacity of a material to absorb and retain moisture from its surroundings is referred to as hygroscopicity [14]. This phenomenon impacts the stability, functionability, and application of a microencapsulated sample [36]. In general, the waste microcapsules absorbed less ambient moisture compared to the waste extract because maltodextrin (the wall material) has a low hygroscopicity, minimizing the hygroscopicity of the extracts [37]. The hygroscopicity values ranged between 20.5 and 23.3 g/100 g for the microencapsulated seed extract and between 11.8 and 13.9 g/100 g for the microencapsulated skin extract, which are shown in Table 2. Similar hygroscopicity values (11.8–19.3 g/100 g) were reported for the microencapsulated peel extract of kinnow using maltodextrin as the wall material and freeze-drying as the microencapsulation method. Worku Dadi et al. [36], studying the hygroscopicity of microencapsulated bioactive products of moringa leaf extract, concluded that the type of wall material (chemical structure/environmental humidity) and microencapsulation method affect this property. Nadali et al. [38] and Tolun et al. [39] reported that maltodextrin with a lower dextrose equivalent (DE) value has lower hygroscopicity since, with a higher DE, the chains are shorter and the glass transition temperature is higher, increasing the hydrophilic groups.

Solubility refers to the ability of a substance to dissolve in a particular solvent, in this case, the microcapsules in water. This property of microcapsules is a critical aspect that can influence the final quality of the reconstituted product [40]. The percentage solubility in water (%) for the microcapsules at different concentrations of maltodextrin for the seed and skin extracts was evaluated (Table 2). The percentage of solubility in water from waste microcapsules is over 4% (Table 2). So, the highest value of solubility in water has been 20.7 ± 0.7%, while the lowest solubility has been 4.3 ± 0.6% for microencapsulated seed extract with MD10 and MD30, respectively. In previous studies, a solubility of 9.1% was reported for microencapsulated betabel extract obtained with maltodextrin [17], whereas microencapsulated cactus pear extract with soluble fiber exhibited a solubility < 1% [41]. On the other hand, a decrease in solubility was observed in both microencapsulated extracts (seed and skin) with increasing maltodextrin concentrations. This behavior exhibited by maltodextrin could be explained by the decrease in the hydrophilic groups available to bind water, which could be occupied with linking bioactive compounds [17].

An important parameter to evaluate is the color of the microcapsules because it could influence their applications and functionality. Therefore, Table 2 presents the results of the color measurement before and after the microencapsulation from waste extracts. The color parameters of the waste extracts were significantly affected by the freeze-drying microencapsulation. The samples were in the first quadrant of the CIELAB space, and it was observed that the effect of adding maltodextrin as the wall material increased the lightness (L*) and the a* and b* coordinates when compared to the non-microencapsulated extract. The b* values of the microencapsulated skin extract significantly increased (*p* < 0.05), showing a tendency toward a yellow color, while the microencapsulated seed extracts with maltodextrin at different concentrations were classified as reddish color (Table 2). Therefore, these changes in color can be related with the color of wall material used (maltodextrin) for the microcapsule’s preparation. A similar behavior was also observed by Ribeiro et al. [32]. The color of the microcapsules with the lowest maltodextrin content had the lowest lightness, whereas the color of the microcapsules with the highest maltodextrin content had the highest lightness, implying that microcapsules with lower maltodextrin contents have more intense color (a* and b* coordinates).

The morphologies of the papaya waste microcapsules were studied using a scanning electron microscope (SEM). The images (Figure 2A(a–f)) are consistent with the broken glass structure observed for microcapsules obtained using freeze-drying, where this irregular shape could protect the bioactive compounds. Similar morphology was observed in freeze-dried microcapsules of lemon [14], juçara palm [42], and mangosteen peel [43] using maltodextrin as a wall material.

The microcapsule morphologies of seed and skin extracts are typically glassy structures with a range size between 242.3 ± 24.3 and 538.9 ± 16.2 µm (Table 2) for different maltodextrin concentrations. It is worth noting that microcapsules typically range in size from 1 to 1000 µm and can be used as multi-unit drug delivery devices (physiological and pharmacokinetics) [44].

According to Figure 2B, the same behavior was observed in the microcapsules of Chilean papaya waste (seed and skin) produced using the freeze-drying technique with maltodextrin at various concentrations (M10, M20, and M30). As demonstrated in Figure 2B, weight losses occurred in three typical temperature ranges for all microcapsules evaluated. The first temperature range (below 200 °C) is associated with weight loss as a result of the sample’s residual water evaporating, or maybe because the microcapsules rehydrated during storage [31,45,46]. Between the temperatures of 200 and 350 °C, the second weight loss associated with the degradation of the wall material and phenolic compounds that were entrapped in the wall material can be observed [31,46]. These results were consistent with maltodextrin degradation at a temperature of about 300 °C (Figure 2B). The third temperature range (above 350 °C) was associated with weight loss due to the decomposition of all sections of the core material (bioactive extract of seed and skin Chilean papaya). A similar behavior was observed for the moringa oleifera leaf powder extract encapsulated in maltodextrin [31]. In general, the microencapsulated seed and skin extracts of Chilean papaya presented significant thermal stability at temperatures less than 200 °C, temperatures widely used in food processes [46].

### 3.3. Phenolic Profile, TPC, TFC, and MEE

The polyphenolic profile of the papaya waste extract and microencapsulated extract (seed and skin) was determined using LC–MS/MS (see Table 3). Based on comparisons with the 21 standards, the polyphenolic profile consists of two groups: flavonoids (rutin and quercetin) and phenolic acids (gallic, ferulic, coumaric, chlorogenic, and caffeic acid), so the contribution of these two groups to each sample was calculated as a contribution percentage (Figure 3). The contribution percentage of the flavonoid group was higher than that of the phenolic acid group in all samples. It should be noted that the phenolic profile varied following processing (microencapsulation), most likely due to structural changes that facilitated the release or degradation of the phenolic groups [47]. Furthermore, the wall materials act as physical barriers, reducing the impacts of oxygen, light, heat, and moisture on the microencapsulated extracts [48].

This study is the first report of the polyphenolic profile in Chilean papaya waste (seed and skin). However, gallic acid, ferulic acid, coumaric acid, chlorogenic acid, caffeic acid, rutin, and quercetin compounds have been detected in the pulp of Chilean papaya [2,49,50]. These compounds have been associated with multiple beneficial properties for human health, for example, antimicrobial [51,52,53,54], antidiabetic [53,55,56], anti-inflammatory [53,54,57], anticancer [53,58], and others [54,59,60,61].

The TPC and TFC results of the seed and skin extracts of Chilean papaya before and after microencapsulation are shown in Table 4. The TPC and TFC were examined against the microencapsulation efficiency (% MEE) from the microencapsulated seed and skin extracts (Table 4). The results showed that the wall material at MD20 exhibited the highest MEE of the TPC and TFC for the microencapsulated seed extract, followed by M30 and M10. However, there was no significant difference (*p* > 0.05) in the MEE between M30 and M10. On the other hand, Table 4 shows that the MEE of the TPC and TFC from the microencapsulated skin extract was influenced by the maltodextrin concentration. When the maltodextrin concentration increased from 10% to 30%, a lower MEE was obtained. This might be attributed to increased quantities of TPC and TFC on the surface, indicating that the compounds were not microencapsulated or adhered to the microcapsule surface [62].

### 3.4. Antioxidant Capacity and Intracellular ROS Generation

Table 4 shows the antioxidant capacity of the extracts (non-microencapsulated) and microcapsules produced by freeze-drying the seed and skin extracts of Chilean papaya and using maltodextrin as the wall material at different concentrations. The results of the antioxidant capacity tests using DPPH, ABTS, and FRAP showed a similar trend for both the papaya seed extract and microencapsulated seed extract, with the following decreasing order: extract > MD20 > MD30 > MD10. On the other hand, for the skin samples, the decreasing order was as follows: extract > MD10 > MD30 > MD20. The results obtained in this study were comparable to those previously reported for Chilean papaya seeds [1]. However, no previous studies had evaluated the antioxidant capacity of Chilean papaya skin.

In general, the antioxidant capacity values in the papaya seed and skin extracts significantly decreased after microencapsulation. The final results demonstrate a lower antioxidant capacity as a result of removing maltodextrin before assays (DPPH, ABTS, and FRAP), with weight considered a previous factor affecting the release of the bioactive extract (core) [14]. Sharayei et al. [63] showed a similar result for antioxidant capacity, indicating that the microcapsules of pomegranate peel extract presented a lower total concentration than the non-microencapsulated extract.

When ROS levels are high, they can either sensitize cancer cells to chemotherapy by increasing oxidative stress or disrupt the cell balance when ROS levels are excessive. These ROS-induced alterations may inhibit tumor cell growth [64]. Previous research has linked oxidative stress and apoptosis to renal illness because they can promote inflammation, tissue damage, fibrosis, and the development of chronic kidney disease [65]. Superoxide and hydroxyl radicals, the main elements of ROS, are produced by H_2_O_2_, which has been widely employed to cause oxidative stress in vitro [66]. Thus, in this study, we evaluated whether the extract and microencapsulated extract of Chilean papaya seed and skin had protective effects against H_2_O_2_-induced apoptosis in Hek293 cells.

The effect of the extracts and microencapsulated seed and skin extracts of Chilean papaya on H_2_O_2_-induced ROS generation in Hek293 cells was measured (Figure 4). It was observed that ROS generation significantly decreased following treatment of the cells with papaya waste extracts. Likewise, both the nuclei of the microcapsules (seed and skin extract) demonstrated a significant reduction in ROS formation.

Recent studies have investigated the protective effects of polyphenols against oxidative stress. For example, Wang et al. [67] demonstrated that phlorizin, a major phenolic compound found in apples, effectively reduced oxidative stress by decreasing ROS production in H_2_O_2_-induced HepG2 cells. Likewise, Wang et al. [68] indicated that polyphenol pretreatment (chlorogenic acid, dihydromyricetin, apigenin, and phloretin) reduces intracellular ROS production of AML-12 and IEC-6 cells.

### 3.5. Antimicrobial Activity

As shown in Figure 5A, both the seed extract and microencapsulated seed extract of Chilean papaya inhibited the growth of all analyzed bacteria, with an inhibition zone > 9 mm. The largest inhibition zone (Figure 5B) was observed for the seed extract against Staphylococcus aureus (22.67 ± 0.58 mm). The MIC value in Figure 5C in the MIC assay was 0.03 ± 0.00 mg/mL. The microencapsulated extract of papaya seed and skin at different maltodextrin concentrations exhibited significantly lower antimicrobial activity against the three evaluated bacteria when compared to the non-microencapsulated extract. However, there was no significant difference (*p* > 0.05) in the antimicrobial activity between MD10, MD20, and M30. This could be explained by the inclusion of maltodextrin in the initial weighing, which is later eliminated before the assay. This inclusion alters the actual concentration of the extract [14]. On the other hand, the skin extract and the microencapsulated skin extract did not present antimicrobial activity for any of the assays evaluated.

Recently, Vega-Gálvez et al. [47] observed a lower antimicrobial effect of the ethanolic extract obtained from Chilean papaya pulp on three bacteria, including *E. coli* and *S. aureus* (inhibition zone between 10.2 and 9.42 mm). Furthermore, higher MIC values (above 200 mg/mL) were observed for the two bacteria when compared to the seed and skin extracts as well as the microencapsulated extracts. Uribe et al. [69] reported similar inhibition zone values (10.13 mm for *E. coli* and 9.57 mm for *S. aureus*) for methanolic pulp extract from Chilean papaya, but they noted that the polar fractions from papaya extracts exhibit higher antimicrobial activity than nonpolar fractions. Egbuonu et al. [70] reported the antimicrobial activity (inhibition zone diameter in mm) of an aqueous extract of Carica papaya peels and seeds at a concentration of 100 mg/mL. They observed values of 12.33 mm and 15.67 mm for *E. coli*, and 8.00 mm and 11.67 mm for *S. aureus*, respectively.

There are several studies on the effects of polyphenols on the antimicrobial activity, where it has been demonstrated that these compounds have different antimicrobial mechanisms [71,72]. According to Ivanov et al. [73], the most promising polyphenols investigated based on their antimicrobial capacity include hesperetin, hesperidin, naringenin, naringin, taxifolin, rutin, isoquercitrin, morin, chlorogenic acid, ferulic acid, p-coumaric acid, and gallic acid. Among them, rutin emerged as the most active representative in inhibiting the formation of bacterial biofilms in vitro. Likewise, Orhan et al. [74] indicated that rutin shows an antimicrobial effect on Staphylococcus aureus, Acinetobacter baumannii, and Pseudomonas aeruginosa. Therefore, rutin, one of the most abundant phenolic compounds found in the seed extract, could contribute to the antimicrobial activity against various bacteria.

## 4. Conclusions

This research successfully microencapsulated bioactive extracts from Chilean papaya waste, particularly the seeds and skin, using maltodextrin as the wall material. The process achieved a high efficiency and extraction yield of bioactive compounds. The microencapsulated seed and skin extracts with MD20 and MD10 exhibited the highest phenolic and flavonoid contents, respectively. Additionally, these extracts showed significant antioxidant capacities and the ability to decrease intracellular ROS levels. In the case of the seed samples, the antimicrobial potential was particularly effective against *S. aureus*. Overall, these findings highlight the potential of utilizing Chilean papaya waste as a valuable resource with promising health benefits and applications in food and pharmaceutical industries.

## Figures and Tables

**Figure 1 antioxidants-12-01900-f001:**
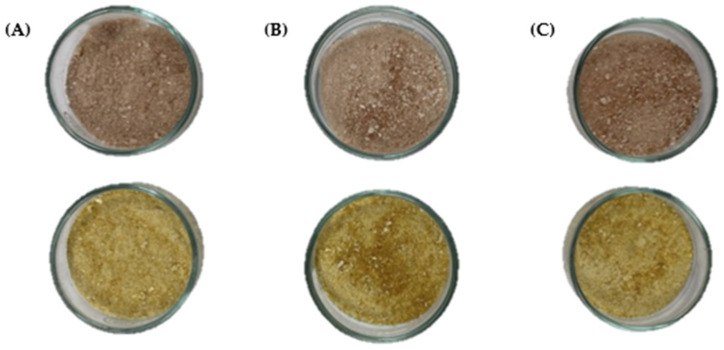
Freeze-dried microcapsules of seed (top of image) and skin (bottom of image) extracts from Chilean papaya at different maltodextrin concentrations: (**A**) MD10 (10%), (**B**) MD20 (20%), and (**C**) MD30 (30%).

**Figure 2 antioxidants-12-01900-f002:**
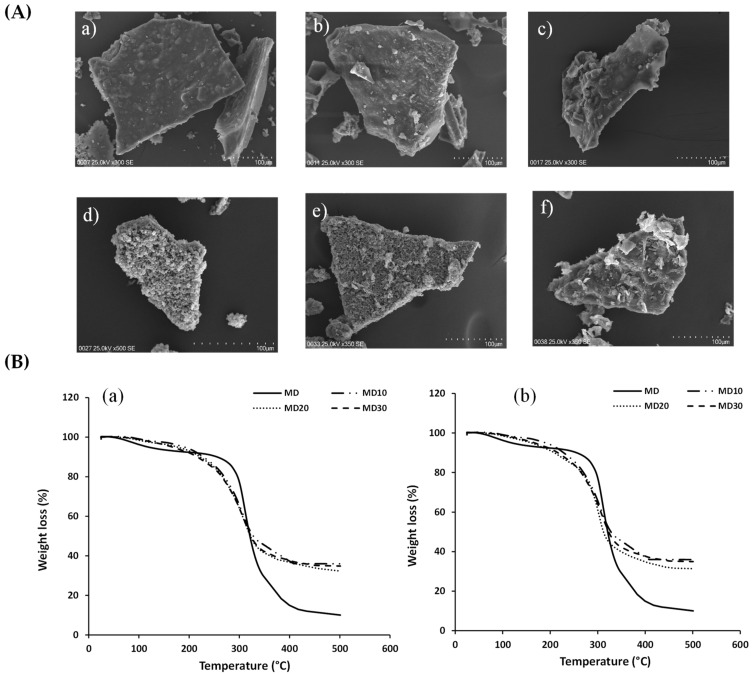
(**A**) SEM images of the papaya waste microcapsules with maltodextrin at 10% (MD10), 20% (MD20), and 30% (MD30) from (**a**–**c**) seed extracts and (**d**–**f**) skin extracts, respectively; and (**B**) thermo-gravimetric analysis (TGA) of microencapsulated seed (**a**) and skin (**b**) extracts produced using the freeze-drying technique with maltodextrin at different concentrations.

**Figure 3 antioxidants-12-01900-f003:**
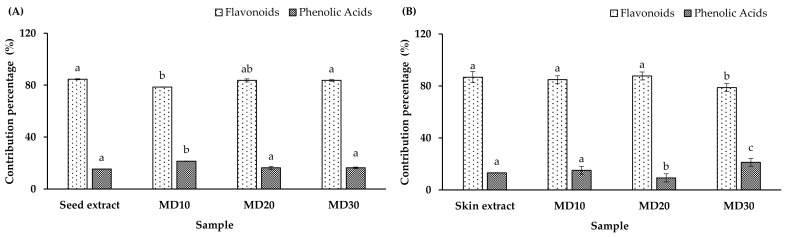
Contribution percentage of phenolic groups obtained using LC–MS/MS analysis: (**A**) seed extract and microencapsulated seed extract and (**B**) skin extract and microencapsulated skin extract. Different lowercase letter in the same bars indicate significant differences between the mean values (*p* < 0.05).

**Figure 4 antioxidants-12-01900-f004:**
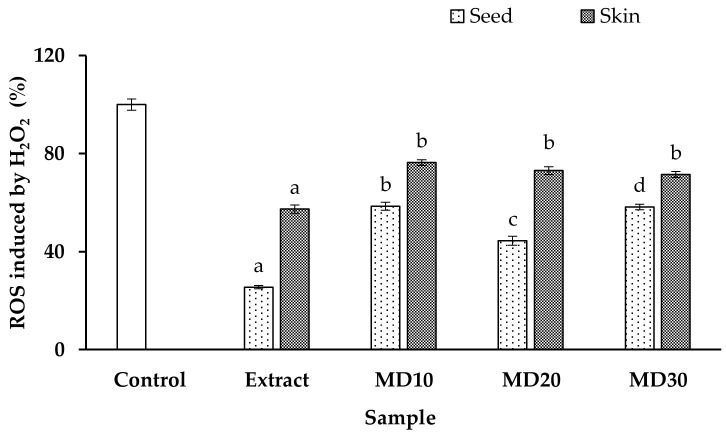
Effects of extracts and microencapsulated seed and skin extracts of Chilean papaya on the production of intracellular reactive oxygen species (ROS). Hek293 cells were exposed to 1 mg/mL of samples for 24 h. Different lowercase letters in the same bars indicate the significant differences from the control (*p* < 0.05).

**Figure 5 antioxidants-12-01900-f005:**
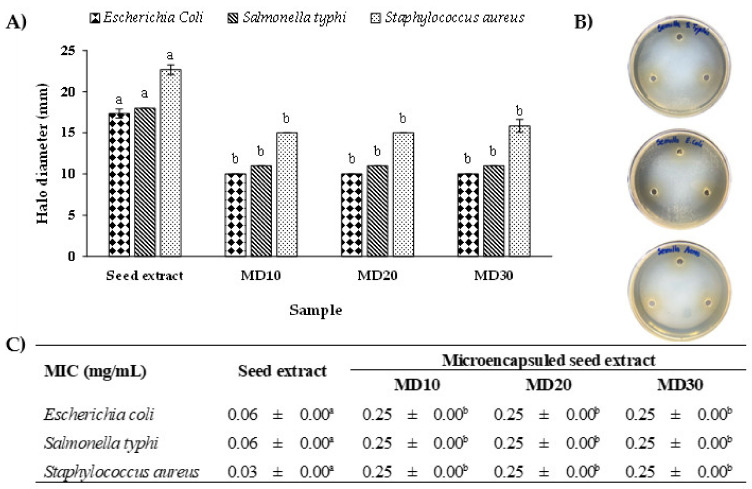
Effect of seed extract and microencapsulated seed extract (1 mg/mL) obtained from Chilean papaya on antimicrobial activity using two assays: (**A**) inhibition zone (diameter, mm) and (**B**) agar diffusion results of seed extract. (**C**) Table with results of MIC values (MIC: minimal inhibitory concentration, mg/mL). Values with different letters in the same bar or row are significantly different (*p* < 0.05).

**Table 1 antioxidants-12-01900-t001:** Proximate analysis from Chilean papaya waste.

Parameters	Content (%)	
	Seed	Skin
Moisture	63.71	90.32
Protein	10.92	1.65
Lipid	11.16	0.20
Fiber total	8.33	1.82
Ash	2.02	1.22
Carbohydrate	12.19	6.61

**Table 2 antioxidants-12-01900-t002:** Physicochemical parameters of extracts and microencapsulated extracts from Chilean papaya waste (mean ± standard error).

Parameters		Microencapsulated Seed Extract				Microencapsulated Skin Extract		
	Seed Extract	MD10	MD20	MD30	Skin Extract	MD10	MD20	MD30
Moisture content (g/100 g)	22.9 ± 1.2 ^a^	9.3 ± 0.1 ^b^	7.0 ± 0.2 ^c^	6.2 ± 1.2 ^c^	23.8 ± 0.1 ^a^	9.9 ± 0.0 ^b^	9.0 ± 0.7 ^b^	4.9 ± 0.2 ^c^
Aw (Dimensionless)	0.517 ± 0.011 ^a^	0.368 ± 0.012 ^b^	0.353 ± 0.005 ^b^	0.297 ± 0.030 ^c^	0.503 ± 0.004 ^a^	0.334 ± 0.005 ^b^	0.282 ± 0.003 ^c^	0.282 ± 0.003 ^c^
Hygroscopicity (g/100 g)	34.2 ± 1.8 ^a^	23.3 ± 1.2 ^b^	22.5 ± 2.0 ^bc^	20.5 ± 1.4 ^c^	31.5 ± 2.4 ^a^	11.8 ± 1.8 ^b^	13.9 ± 0.9 ^b^	13.1 ± 1.5 ^b^
Solubility in water (%)	5.3 ± 0.0 ^a^	20.7 ± 0.7 ^b^	13.4 ± 0.5 ^c^	4.3 ± 0.6 ^d^	9.6 ± 0.1 ^a^	20.3 ± 0.6 ^b^	19.1 ± 0.9 ^b^	16.1 ± 0.2 ^c^
Lightness (L*)	24.7 ± 1.1 ^a^	45.3 ± 0.7 ^b^	52.0 ± 0.1 ^c^	60.4 ± 0.1 ^d^	31.1 ± 1.3 ^a^	78.6 ± 0.7 ^b^	68.1 ± 1.0 ^c^	66.9 ± 0.2 ^c^
a* color coordinate	7.9 ± 0.2 ^a^	10.9 ± 1.2 ^b^	11.0 ± 0.9 ^b^	13.6 ± 0.8 ^c^	7.2 ± 0.2 ^a^	6.8 ± 0.3 ^a^	11.5 ± 0.1 ^b^	13.1 ± 1.1 ^c^
b* color coordinate	11.6 ± 0.1 ^a^	13.5 ± 0.3 ^b^	13.2 ± 0.3 ^b^	18.8 ± 0.0 ^c^	25.8 ± 0.4 ^a^	33.6 ± 0.7 ^b^	41.8 ± 0.4 ^c^	47.7 ± 0.5 ^d^
Size of microcapsule (µm)	-	243.5 ± 15.9	242.3 ± 24.3	328.8 ± 22.8	-	247.8 ± 23.5	538.9 ± 16.2	537.3 ± 12.7

Different lowercase letters in rows indicate significant differences between the mean values (*p* < 0.05) for each sample (microencapsulated seed and skin extracts; *n* = 3).

**Table 3 antioxidants-12-01900-t003:** Phenolic profile of extracts and microencapsulated extracts from Chilean papaya waste (mean ± standard error).

Phenolic Profile (µg/mL)	Extract	Microencapsulated Extract		
		MD10	MD20	MD30
	Seed Samples			
Gallic acid	171.0 ± 0.4 ^a^	18.6 ± 0.1 ^b^	7.1 ± 0.1 ^c^	32.2 ± 0.1 ^d^
Ferulic acid	4.1 ± 0.1 ^a^	8.0 ± 0.1 ^b^	5.1 ± 0.1 ^c^	4.0 ± 0.1 ^d^
Chlorogenic acid	159.2 ± 0.3 ^a^	558.6 ± 0.9 ^b^	398.6 ± 0.2 ^c^	334.1 ± 0.3 ^d^
Caffeic acid	7.8 ± 0.2 ^a^	ND	ND	ND
Coumaric acid	12.3 ± 0.1 ^a^	8.4 ± 0.1 ^b^	20.0 ± 0.2 ^c^	20.1 ± 0.1 ^c^
Rutin	1878.6 ± 0.8 ^a^	2093.7 ± 0.8 ^b^	2126.4 ± 0.9 ^c^	1927.0 ± 0.9 ^d^
Quercetin	67.9 ± 0.2 ^a^	79.2 ± 0.2 ^b^	76.2 ± 0.4 ^c^	69.1 ± 0.2 ^d^
	Skin Samples			
Gallic acid	21.0 ± 0.2 ^a^	26.8 ± 0.1 ^b^	13.4 ± 0.1 ^c^	3.0 ± 0.1 ^d^
Ferulic acid	7.3 ± 0.1 ^a^	6.6 ± 0.1 ^b^	4.9 ± 0.1 ^c^	5.1 ± 0.1 ^c^
Chlorogenic acid	194.5 ± 0.5 ^a^	321.2 ± 0.2 ^b^	180.5 ± 0.2 ^c^	648.5 ± 0.5 ^d^
Caffeic acid	35.5 ± 0.2 ^a^	ND	ND	ND
Coumaric acid	7.0 ± 0.1 ^a^	15.3 ± 0.1 ^b^	12.2 ± 0.3 ^c^	10.0 ± 0.1 ^d^
Rutin	1682.7 ± 0.9 ^a^	2114.4 ± 0.8 ^b^	2013.1 ± 0.9 ^c^	2016.7 ± 0.9 ^d^
Quercetin	63.6 ± 0.3 ^a^	70.0 ± 0.2 ^b^	66.4 ± 0.7 ^c^	92.7 ± 0.6 ^d^

Different lowercase letters in rows indicate significant differences between the mean values (*p* < 0.05) for each sample (microencapsulated seed and skin extracts; *n* = 3). ND: not detected at the conditions tested.

**Table 4 antioxidants-12-01900-t004:** Bioactive compounds, microencapsulation efficiency (MEE), and antioxidant capacity (A) of extract and microencapsulated extract from Chilean papaya waste (mean ± standard error).

Assay		Microencapsulated Seed Extract				Microencapsulated Skin Extract		
	Seed Extract	MD10	MD20	MD30	Skin Extract	MD10	MD20	MD30
TPC (mg/g DW)	44.49 ± 1.68 ^a^	41.11 ± 3.32 ^a^	44.20 ± 0.50 ^a^	32.85 ± 3.00 ^b^	29.24 ± 1.99 ^a^	16.06 ± 0.82 ^b^	11.69 ± 1.13 ^c^	10.28 ± 1.89 ^c^
Surface TPC (mg/g DW)	-	4.91 ± 0.48 ^a^	3.46 ± 0.26 ^b^	3.66 ± 0.22 ^b^	-	2.76 ± 0.10 ^a^	3.12 ± 0.17 ^b^	3.52 ± 0.21 ^c^
TFC (mg/g DW)	1.02 ± 0.01 ^a^	0.89 ± 0.06 ^b^	0.90 ± 0.04 ^b^	0.44 ± 0.04 ^c^	6.41 ± 0.03 ^a^	1.05 ± 0.08 ^b^	1.32 ± 0.07 ^c^	0.75 ± 0.07 ^d^
Surface TFC (mg/g DW)	-	0.08 ± 0.01 ^a^	0.02 ± 0.00 ^b^	0.01 ± 0.00 ^c^	-	0.05 ± 0.00 ^a^	0.09 ± 0.00 ^b^	0.05 ± 0.00 ^c^
MEE of TPC (%)	-	86.61 ± 2.83 ^a^	92.24 ± 0.53 ^b^	88.84 ± 0.39 ^a^	-	82.80 ± 0.74 ^a^	71.06 ± 0.26 ^b^	68.91 ± 2.71 ^b^
MEE of TFC (%)	-	90.85 ± 0.12 ^a^	98.00 ± 0.08 ^b^	97.02 ± 0.29 ^b^	-	95.53 ± 0.12 ^a^	93.36 ± 0.21 ^b^	92.70 ± 0.15 ^b^
A_DPPH_ (µmol/g DW)	12.06 ± 0.11 ^a^	8.23 ± 0.08 ^b^	12.12 ± 0.15 ^a^	10.05 ± 0.2 ^c^	11.45 ± 0.10 ^a^	10.45 ± 0.14 ^b^	7.67 ± 0.23 ^c^	9.60 ± 0.46 ^d^
A_ABTS_ (µmol/g DW)	258.38 ± 1.33 ^a^	148.31 ± 1.64 ^b^	203.50 ± 4.06 ^c^	176.82 ± 7.93 ^d^	270.46 ± 1.35 ^a^	221.89 ± 7.91 ^b^	110.81 ± 5.60 ^c^	163.03 ± 5.29 ^d^
A_FRAP_ (µmol/g DW)	271.89 ± 4.97 ^a^	208.71 ± 6.21 ^b^	236.34 ± 4.11 ^c^	212.01 ± 7.23 ^b^	206.45 ± 8.50 ^a^	163.70 ± 8.46 ^b^	138.24 ± 6.79 ^c^	163.63 ± 6.90 ^b^

Different lowercase letters in rows indicate significant differences between the mean values (*p* < 0.05) for each sample (microencapsulated seed and skin extracts; *n* = 3). TPC: total phenolic content; TFC: total flavonoid content; MEE: microencapsulation efficiency; A_DPPH_: antioxidant capacity determined using a DPPH radical scavenging assay; A_ABTS_: antioxidant capacity determined using ABTS radical scavenging assay; A_FRAP_: antioxidant capacity determined using ferric reduction antioxidant power assay.

## Data Availability

Not applicable.

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
