# Peer review of "Microencapsulation of Chilean Papaya Waste Extract and Its Impact on Physicochemical and Bioactive Properties"

_antioxidants, 2023, doi:10.3390/antiox12101900_

Round 1

Reviewer 1 Report

I found the manuscritpt on microencapsulation of chilean papaya waste extract really interesting and well structured.

The following are my observations:

abstract (line 23) please change "Seed" with "seed" 

I would suggest changing the last part of the introduction and not reporting the objectives as a simple list and try to "stress" the importance and need for encapsulation of these polyphenolic extracts.

section 2.2. I would suggest to add a table in which all data of raw materials are reported. Please change , with . when numbers are added. 

section 2.8 please add the doses used to perform the assays of total phenolic and flavonoid contents. 

section 2.10 please add the information about gradient 

section 2.11 please add the information about doses of extracts and the method used for testing microencapsulated extracts

section 2.12 please change 5x104 with 5x104

Figure 1 letters a) b) and c) should be added 

line 283 it should be Figure 2 and  not Figure 3

Why the antimicrobial results of skin extract and microencapsulated ones are not reported? 

line 435 please change S. aureus with S.aureus  

Author Response

Comments Reviewer 1

We appreciate the Reviewer's valuable input and their commitment to improving our work. We have diligently addressed the points raised and have incorporated the necessary changes, as outlined below:

Comment 1

abstract (line 23) please change "Seed" with "seed" 

Answer 1

Line 23- - Thanks for the correction; the “Seed" was changed to "seed"

Comment 2

I would suggest changing the last part of the introduction and not reporting the objectives as a simple list and try to "stress" the importance and need for encapsulation of these polyphenolic extracts.

Answer 2

Thanks for the suggestion. The last part of the introduction has been improved as per your request in the current manuscript review:

“In this research, we investigated the microencapsulation of bioactive extracts derived from Chilean papaya waste, which includes both the seed and skin components. Our primary objectives encompassed not only the production of bioactive microcapsules from papaya waste extracts but also an extensive analysis of their physicochemical properties. We assessed their total antioxidant capacity (TAC) using various methods, including DPPH, FRAP, ABTS, and conducted kinetic analysis of intracellular ROS generation. Additionally, we determined the total polyphenol content (TPC) and total flavonoid content (TFC). To further enhance our understanding, we identified and quantified the phenolic compounds through LC-MS/MS analysis. The findings from this research open promising possibilities for the utilization of fruit waste in various industries, including but not limited to food, cosmetics, and pharmaceuticals, thereby highlighting its potential to transform discarded resources into valuable assets.”

Comment 3

section 2.2. I would suggest to add a table in which all data of raw materials are reported. Please change, with . when numbers are added. 

Answer 3

Thanks for the suggestion. The table and its mention in section 2.2 have been added to the manuscript:

“Table 1 presents the proximate analysis of the raw materials (seed and skin) obtained from Chilean papaya waste.”

Table 1. Proximate analysis from Chilean papaya waste.

Parameters

Content (%)

Seed

Skin

Moisture

63.71

90.32

Protein

10.92

1.65

Lipid

11.16

0.20

Fiber total

8.33

1.82

 Ash

2.02

1.22

Carbohydrate

12.19

6.61

Comment 4

section 2.8 please add the doses used to perform the assays of total phenolic and flavonoid contents. 

Answer 4

In Section 2.8 of the manuscript, significant enhancements have been made to the TPC and TFC methodologies:

“To determine TPC and TFC in microcapsules, we followed the methodology proposed by Saikia et al. [19] with some modifications. We took 100 mg of the microencapsulated sample and added 1 mL of a mixture containing ethanol, acetic acid, and water in a ratio of 50:8:42 (v/v/v). This mixture was vortexed for 1 minute at room temperature, followed by 20 minutes using a water bath sonicator (VWR® Symphony™ Ultrasonic Cleaner, 45 kHz operation frequency). The resulting supernatant was then centrifuged (ORTO ALRESA, Bioprocen 22 R model) for 5 minutes at 112 g and filtered. Finally, the supernatant obtained after filtration was subjected to sample concentration (ALLSHENG model MD200-1A) until complete drying, and the volume was reconstituted with 70% ethanol.

The TPC was determined using the Folin-Ciocalteu (FC) methodology [20]. Specifically, 20 µL of the extract was mixed with 100 µL of the FC reagent at a 1:10 (v/v) ratio and combined with 80 µL of a 7.5% sodium carbonate solution. The resulting mixture was then allowed to react for 40 minutes in darkness before measuring the absorbance at 765 nm (ALLSHENG FlexA-200 spectrophotometer). The TPC was calculated using a calibration curve with gallic acid (GA) as the standard and expressed in milligrams of gallic acid equivalent (GAE) per gram of dry weight of the sample (D.W).

The Total Flavonoid Content (TFC) was determined using the aluminum chloride procedure [21]. To accomplish this, 100 microliters of the samples were mixed with 100 µl of a 2% AlCl3 ethanol solution and incubated for 30 min at room temperature. Subsequently, the absorbance was measured at 420 nm using an ALLSHENG FlexA-200 spectrophotometer. TFC was calculated using a calibration curve with quercetin as the standard and expressed in milligrams of quercetin equivalent (QE) per gram of dry weight of the sample (DW).”

Comment 5

section 2.10 please add the information about gradient.

Answer 5

In Section 2.10 of the manuscript, the information about gradient was added:

“The gradient was programmed as follows: 0 – 1 min, 15% B; 1 – 17 min, 15-100 % B; 17-21 min 100-100% B; 21-22 min, 100-15 % B; 22-25 min, 15-15% B.”

Comment 6

section 2.11 please add the information about doses of extracts and the method used for testing microencapsulated extracts.

Answer 6

In Section 2.11 of the manuscript, significant enhancements have been made to the Antioxidant capacity methodologies:

“For each measurement, we combined 50 µL of the extract with 150 µL of the respective solution (DPPH, ABTS, or FRAP). After a 30-minute incubation period, we recorded the absorbance at 517 nm for DPPH, at 732 nm for ABTS, and at 593 nm for FRAP, using the ALLSHENG model FlexA-200 multi-plate reader. Trolox served as our calibration standard. The results are reported as µmol of Trolox equivalents (ET) per gram of dry weight of the sample (DW).”

Comment 7

section 2.12 please change 5x104 with 5x104

Answer 7

Section 2.12- The superscript was corrected in the Materials and Methods section.

Comment 8

Figure 1 letters a) b) and c) should be added 

Answer 8

Figure 1- Thanks for the correction, the letters were added.

Comment 9

line 283 it should be Figure 2 and  not Figure 3

Answer 9

Line 283- Thank you for the correction; the text has been changed.

Comment 10

Why the antimicrobial results of skin extract and microencapsulated ones are not reported? 

Answer 10

We did not report the antimicrobial results for both the skin extract and the microencapsulated samples because none of the evaluated assays showed any antimicrobial activity in either of them.

This information can be found in the manuscript on lines 432-433.

Comment 11

line 435 please change S. aureus with S.aureus  

Answer 11

Line 435- Thank you for the correction; the text has been changed.

Reviewer 2 Report

This paper describes the microencapsulation of Chilean papaya waste extract and its impact on physicochemical and bioactive properties. The article is quite complete, it is of interest to the scientific community, the methods and statistics used are appropriate and the results and discussion are conveniently described. The work is well discussed and is supported by the references provided by the authors. The English language is correct. The work is interesting and delves in the utilization of wate from papaya.

I consider that the article is appropriate to be published in Antioxidants journal once the authors have made modifications to it.

-          Title: Capitalize each word according the format of the journal.

-          Lines 18, 203, 228, 243, ,…: Put a separation after and before “±”, “=”, “<”,. Unify and apply to the entire document.

-          Lines 70, 92, 118, 141, 142, 175,……: Include the city and country of all the companies cited, and cite the companies of all the reagents and equipment’s employed. In case of USA companies, include the city and the state abbreviation. Unify and apply to the entire document.

-          Lines 79, 89, 97, 104, ……: Capitalize each word according the format of the journal. Unify and apply to the entire document.

-          Lines 91, 159, ….: Use “mL” instead of “ml”. Unify and apply to the entire document.

-          Lines 94,…..: Rpm: Use “g” or include the orbital radius.

-          Lines 94,…..: Use “min” instead of “minutes”. Unify and apply to the entire document.

-          Lines 103,143,  167, 279, …..: Put a separation between a number and “ºC”. Unify and apply to the entire document.

-          Section 2.8: Include calibration curve, r2 and range of linearity of gallic acid.

-          Line 145: “0.5”.

-          Section 2.10: Describe the identified flavonoids (masses, fragmentations, retention time, etc.). Include mass spectrum of each of these flavonoids.

-          Section 2.11: Include calibration curve, r2 and range of linearity of trolox.

-          Line 164: “1 µL”.

-          Lines 185, 187, 243, 260, 261, 305,…..: Put “p” and “n” in italics. Unify and apply to the entire document.

-          Lines 212, 213, Table 1, ,….: Put a separation between the number and the units. Unify and apply to the entire document.

-          Lines 263, 265, Table 1, …: “µm”.

-          Line 285: Put “Moringa oleifera” in italics.

-          Line 377: In “H2O2” put “2” in subscript.

-          Figure 5: “Escherichia coli”.

-          References 4, 8, ….: Information is missing or the format is not the correct. Revise all the references.

-           

Moderate editing of English language required

Author Response

Comments Reviewer 2

We extend our gratitude to the Reviewer for their keen interest in our research and for providing constructive feedback aimed at enhancing the quality of our manuscript. We have taken great care to address each of the Reviewer's comments, and we've made the appropriate revisions, as detailed below:

Comment 1

Title: Capitalize each word according the format of the journal.

Answer 1

Title - Thanks for the correction; the title was modified according to format of the Journal:

“Microencapsulation of Chilean Papaya Waste Extract and Its Impact on Physicochemical and Bioactive Properties.”

Comment 2

Lines 18, 203, 228, 243, ,…: Put a separation after and before “±”, “=”, “<”,. Unify and apply to the entire document.

Answer 2

We have added a space before and after “±”, “=” and “<”. These changes have been applied uniformly throughout the entire document.

 Comment 3

Lines 70, 92, 118, 141, 142, 175,……: Include the city and country of all the companies cited, and cite the companies of all the reagents and equipment’s employed. In case of USA companies, include the city and the state abbreviation. Unify and apply to the entire document.

Answer 3

The task of adding city and country details for cited companies, citing reagent and equipment providers, including city and state (for U.S. companies), has been reviewed and applied in the manuscript.

Comment 4

Lines 79, 89, 97, 104, ……: Capitalize each word according the format of the journal. Unify and apply to the entire document.

Answer 4

Thanks for the correction; all sections have been modified according to the format of the journal.

Comment 5

Lines 91, 159, ….: Use “mL” instead of “ml”. Unify and apply to the entire document.

Answer 5

We have replaced “ml” with “mL”. These modifications have been consistently applied throughout the entire document.

Comment 6

Lines 94,…..: Rpm: Use “g” or include the orbital radius.

Answer 6

We have replaced “Rpm” with “g”. These modifications have been consistently applied throughout the entire document.

 Comment 7

Lines 94,…..: Use “min” instead of “minutes”. Unify and apply to the entire document.

Answer 7

We have substituted “minutes” with “min”. These changes have been uniformly implemented across the entire document.

Comment 8

Lines 103,143,  167, 279, …..: Put a separation between a number and “ºC”. Unify and apply to the entire document.

Answer 8

We have introduced a space between numerical values and 'ºC.' These adjustments have been consistently incorporated throughout the entire document.

Comment 9

Section 2.8: Include calibration curve, r2 and range of linearity of gallic acid.

Answer 9

 In Section 2.8: We have included the calibration curve, R² value, and the range of linearity for GA.

Comment 10

Line 145: “0.5”.

Answer 10

Thank you for the correction; the numerical value has been corrected.

Comment 11

Section 2.10: Describe the identified flavonoids (masses, fragmentations, retention time, etc.). Include mass spectrum of each of these flavonoids.

Answer 11

The mass spectra of the compounds used as standards for polyphenols in the samples have been included in the supplementary material. The data presented is in accordance with our non-targeted identification approach.

 Comment 12

Section 2.11: Include calibration curve, r2 and range of linearity of trolox.

Answer 12

In Section 2.11: We have included the calibration curve, R² value, and the range of linearity for Trolox.

 Comment 13

 Line 164: “1 µL”.

Answer 13

Okay, we have added a space between the number and the unit.

Comment 14

Lines 185, 187, 243, 260, 261, 305,…..: Put “p” and “n” in italics. Unify and apply to the entire document.

Answer 14

We have italicized 'p' and 'n' and applied this change uniformly throughout the entire manuscript.

Comment 15

Lines 212, 213, Table 1, ,….: Put a separation between the number and the units. Unify and apply to the entire document.

Answer 15

We have added a space between the number and the units, and this change has been consistently applied throughout the entire manuscript.

Comment 16

Lines 263, 265, Table 1, …: “µm”.

Answer 16

Thank you for the correction; the unit has been corrected.

Comment 17

Line 285: Put “Moringa oleifera” in italics.

Answer 17

Line 285, we have italicized “Moringa oleifera” as requested.

 Comment 18

Line 377: In “H2O2” put “2” in subscript.

Answer 18

Line 377, we have placed '2' in subscript in “H2O2” as requested.

 Comment 19

Figure 5: “Escherichia coli”.

Answer 19

In Figure 5, we have made the necessary adjustment to “Escherichia coli” as requested.

 Comment 20

References 4, 8, ….: Information is missing or the format is not the correct. Revise all the references.

Answer 20

We have reviewed references and have either added missing information or corrected the format in the manuscript.

Reviewer 3 Report

This research article present results of microencapsulation of Chilean papaya seeds and skin extracts using maltodextrin at different quantities as wall material. Extracts from Chilean papaya seeds and skin were prepared in laboratory conditions, so the question is whether the pomace is the same as that obtained under industrial conditions.

The number of revolutions does not give an indication of the centrifugal force, as it will be different for rotors of different diameters. It is more usual to use g, not rpm.

It is not clear in Figure 2 that it is a and e – not incapsulated particles of the extracts?

Line 282 should be Figure 2, not 3.

Table 2. Phenolic profile of extract and microencapsulated extract from Chilean papaya waste: This table raises many questions. Comparing the results for gallic acid, we can see that it decreases after encapsulation, but it is most abundant in the MD30 sample, which is the one with the highest maltodextrin content. Chlorogenic acid content increases many times in the encapsulated samples, but when comparing the encapsulated samples, the trend is consistent with the increase in maltodextrin content.

There was a lack of justification for the need to encapsulate peel and seed extracts if their antioxidant and antimicrobial properties are better compared to encapsulated extracts. The benefits of mixing with maltodextrin and lyophilisation?

Author Response

Comments Reviewer 3

We appreciate the Reviewer's valuable input and their commitment to improving our work. We have diligently addressed the points raised and have incorporated the necessary changes, as outlined below:

Comment 1

This research article present results of microencapsulation of Chilean papaya seeds and skin extracts using maltodextrin at different quantities as wall material. Extracts from Chilean papaya seeds and skin were prepared in laboratory conditions, so the question is whether the pomace is the same as that obtained under industrial conditions.

Answer 1

The research article presents results of the microencapsulation of Chilean papaya seed and skin extracts using maltodextrin in varying quantities as a coating material. Although these extracts were prepared under laboratory conditions, efforts were made to obtain the raw material (seed and skin waste) in a manner closely resembling the practices of the canning industry, as these operations still employ manual processes to produce preserves.

Comment 2

The number of revolutions does not give an indication of the centrifugal force, as it will be different for rotors of different diameters. It is more usual to use g, not rpm.

Answer 2

We have replaced “Rpm” with “g”. These modifications have been consistently applied throughout the entire document.

Comment 3

It is not clear in Figure 2 that it is a and e – not incapsulated particles of the extracts?

Answer 3

Thank you for your question. The letters 'a' and 'e' in Figure 2 represented images of the lyophilized and pulverized raw material. These letters were removed from the figure to prevent confusion and to focus attention on the encapsulated particles of the extracts, which were the subject of interest in the study.

Comment 4

Line 282 should be Figure 2, not 3.

Answer 4

Line 282- Thank you for the correction; the Figure number has been changed.

Comment 5

Table 2. Phenolic profile of extract and microencapsulated extract from Chilean papaya waste: This table raises many questions. Comparing the results for gallic acid, we can see that it decreases after encapsulation, but it is most abundant in the MD30 sample, which is the one with the highest maltodextrin content. Chlorogenic acid content increases many times in the encapsulated samples, but when comparing the encapsulated samples, the trend is consistent with the increase in maltodextrin content.

Answer 5

Thank you for your question.

The variability in phenolic profiles between extracts and microcapsules can be attributed to several factors. These include differences in microencapsulation efficiency, as some compounds may be lost during the microencapsulation process. The choice of wall material also plays a role, as various wall materials have different affinities for phenolic compounds, affecting their retention. Variations in processing conditions, including the experimental procedure, can lead to differences. Additionally, the sensitivity of phenolic compounds to the microencapsulation process or storage, variations in particle size (smaller particles may release compounds more rapidly), and inherent variability in raw materials contribute to this variability. Therefore, optimizing the microencapsulation process in future work could be of interest to minimize the variability in the phenolic profile.

Comment 6

There was a lack of justification for the need to encapsulate peel and seed extracts if their antioxidant and antimicrobial properties are better compared to encapsulated extracts. The benefits of mixing with maltodextrin and lyophilization?

Answer 6

Thank you for your question regarding the decrease in bioactive properties evaluated in this work. It's important to note that the initial weighing of the microcapsules, which is used to determine bioactive properties, includes the wall material (maltodextrin). However, this wall material is removed before testing. As a result, the microencapsulated waste extracts exhibited lower values compared to the extract. This behavior in terms of bioactive properties has been reported by other authors as well [1,2]. Additionally, it's worth mentioning that the microencapsulation efficiency was found to be over 50%, indicating a successful microencapsulation process.

The choice of microencapsulation of waste extracts with maltodextrin using the freeze-drying method is driven by several reasons. Maltodextrin, serving as a protective matrix, ensures the preservation and stability of bioactive compounds over time. Furthermore, its ability to enhance water solubility guarantees a rapid release upon rehydration. This approach also transforms potentially challenging-to-handle waste extracts into easily manageable powdered forms, providing convenience in various applications (e. g. in the food industry). Moreover, the method offers controlled release properties. Overall, the versatility of microcapsules allows for their integration into a wide array of products, optimizing the usability and market potential of waste extracts across diverse industries.

References

  1. Bernardes, A.L.; Moreira, J.A.; Tostes, M. das G.V.; Costa, N.M.B.; Silva, P.I.; Costa, A.G.V. In vitro bioaccessibility of microencapsulated phenolic compounds of jussara (Euterpe edulis Martius) fruit and application in gelatine model-system. LWT - Food Sci. Technol. 2019, 102, 173–180, doi:10.1016/j.lwt.2018.12.009.
  2. Sharayei, P.; Azarpazhooh, E.; Ramaswamy, H.S. Effect of microencapsulation on antioxidant and antifungal properties of aqueous extract of pomegranate peel. J. Food Sci. Technol. 2020, 57, 723–733, doi:10.1007/s13197-019-04105-w.

Round 2

Reviewer 2 Report

The authors have made the indicated modifications and the article has improved substantially. For this reason, I consider that the article can be considered for publication in Antioxidants journal in its current form.

Minor editing of English language required

Reviewer 3 Report

The authors corrected the manuscript, but its scientific soundness remained moderate